# The KIDSCREEN-27 Quality of Life Measure for Romanian Children Aged 6: Reliability and Validity of the Romanian Version

**DOI:** 10.3390/healthcare10071198

**Published:** 2022-06-27

**Authors:** Viorel Petru Ardelean, Vasile Liviu Andrei, Caius Călin Miuţa, Iuliana Boros-Balint, Grațiela-Flavia Deak, Andor Molnar, Tamás Berki, Ferenc Győri, Vlad Adrian Geantă, Cristina Adriana Dehelean, Florin Borcan

**Affiliations:** 1Faculty of Physical Education and Sport, Aurel Vlaicu University of Arad, 2-4, Elena Dragoi, 310330 Arad, Romania; liviu.andrei@uav.ro (V.L.A.); caius.miuta@uav.ro (C.C.M.); 2Faculty of Physical Education and Sport, Babeș-Bolyai University of Cluj-Napoca, 7 Pandurilor Str., 400376 Cluj-Napoca, Romania; iuliana.borosbalint@ubbcluj.ro (I.B.-B.); gratiela.deak@ubbcluj.ro (G.-F.D.); 3Institute of Physical Education and Sport Sciences, Faculty of Education, University of Szeged, Hattyas u. 10, H-6725 Szeged, Hungary; andor.h.molnar@gmail.com (A.M.); berki.tamas.laszlo@szte.hu (T.B.); 4Faculty of Health Sciences, Institute of Physiotherapy and Sport Science, University of Pécs, 4 Vörösmarty u., H-7621 Pécs, Hungary; ferenc.gyori@etk.pte.hu; 5Faculty of Science, Physical Education and Informatics, University of Piteşti, 7 Normal School Alley, 110254 Piteşti, Romania; vladu.geanta@gmail.com; 6Faculty of Pharmacy, “Victor Babes” University of Medicine and Pharmacy Timisoara, 2 Eftimie Murgu Square, 300041 Timisoara, Romania; cadehelean@umft.ro (C.A.D.); fborcan@umft.ro (F.B.)

**Keywords:** health-related quality of life, preparatory school children, wellbeing

## Abstract

The KIDSCREEN-27 represents a standardized, worldwide instrument, employed to assess the health-related quality of life in children. The purpose of the present study is to validate the KIDSCREEN-27 questionnaire for 6-year-old preparatory school children and verify its reliability, as well as to perform a comparison regarding the quality of children’s lives living in two cities in Romania: Arad, a provincial city, versus the second most developed city in the country, Cluj-Napoca. A total of 256 children of 6 years of age, who come from families with both parents, with a medium to high socioeconomic status and a good health status, were included in the analysis, using the KIDSCREEN-27 questionnaire at three assessment time points with a re-test period of two weeks. Results indicated that the KIDSCREEN-27 turned out to be suitable for use in 6-year-old Romanian children. Analysis regarding the psychometric properties showed that the Cronbach’s alpha ranged from 0.554 to 0.661 at the end of the study. The Pearson correlation coefficients showed statistically significant differences between the items of each area investigated. In conclusion, there is a growing need to periodically monitor the health status of children to avoid possible problems which may occur.

## 1. Introduction

Quality of life is a broad general multidimensional concept which encompasses either health or non-health-related objective and subjective aspects of life [1]. For each human being, the subjective aspects of life are the most important, because they are directly responsible for physical, social and individual wellbeing. Mental and physical health, as well as lifespan, are determined by subjective wellbeing, more precisely by life satisfaction, viewed from the perspective of each individual. Optimistic people are the most satisfied with their lives. Moreover, they always have a positive mindset and are mentally built to find favorable answers in different life situations [2]. These people experience high levels of emotions and pleasant moods and are motivated, predicting greater career success and better social relations, and finally have better health [3]. Subjective wellbeing is also determined by good health, healthy lifestyle, hope, a commitment to achieving important goals, a feeling of fulfillment, living condition, resources, education and even personality [4,5]. A healthy lifestyle, which leads to good health and life satisfaction increase, is based on health-related behaviors, such as dieting, physical activity, lack of psychological stress, lack of alcohol, smoking or illicit substances use and sleeping enough [6,7,8,9,10]. Dispositional optimism also contributes to life satisfaction increase. Hope and positive expectations every day make an individual motivated to exert effort, to achieve goals and engagements and to overcome themselves as well as any obstacle [3]. Monitoring the health-related quality of life (HRQL) has a growing significance because it describes health-related parameters such as physical and psychological wellbeing, peer and social acceptance, as well as behavioral aspects defined by emotional or mental experiences [11]. Nima and co-workers have confirmed that subjective wellbeing is well adjusted to the interaction between physiological, psychological and social factors which have a cause–effect relation on health [12].

The quality of life assessment could be considered as an indirect evaluation of mental health status, principally for the population under 18 years of age. A great challenge is the quality of life assessment of children due to the difficulty in evaluating their subjective wellbeing [13]. Continuous monitoring of the everyday emotions and behavior of a child or adolescent, alongside with observation of their mental symptoms, could prevent the extent of serious problematic mental health, even depression [14,15]. For healthy development, children need various necessities that contribute to the formation of subjective wellbeing as a complex of factors [16]. Their biological needs involve negative and positive emotional reactions, safety needs, psychological needs (life satisfaction) and social needs (life harmony and peer relationships). As a child’s quality of life can be assessed by the child’s self-report or the parents’ report, selecting the most appropriate tool is an additional challenge [17].

Worldwide, there are generic and specific instruments suitable for evaluating the quality of life (QL) of children. The generic instrument is suitable for all population subgroups, whereas the specific one is suitable for individuals with disabilities or who suffer from various illnesses [18,19]. The generic instrument proved to be useful for children and adolescents because it identifies that category of children and adolescents who exhibit a risk regarding health problems, thus preventing the aggravation of these problems and the development of any disease or disability [20]. Lately, an increased number of questionnaires addressing children’s self-perspective HRQL have been published [21,22,23,24,25,26]. Each questionnaire is focused on different parameters regarding health-related quality of life, such as life functioning, life satisfaction, disability and health [27].

The KIDSCREEN questionnaire is a generic measure of quality of life for children and adolescents, designed in such a way to cover screening, monitoring and evaluation tools, being suitable in various settings, such as schools, hospitals or even research centers. This instrument of assessment has been developed in 13 European countries, both initially and simultaneously [28]. Globally, the KIDSCREEN instrument is available in three versions, in which KIDSCREEN-52 represent the original version based on 52 items, measuring ten areas of quality of life (physical and psychological wellbeing, emotions and moods, self-perception, autonomy, home life and parent relations, peers and friends’ support, school environment and bullying) [29,30,31]. KIDSCREEN-27 is a multidimensional scale which include 27 self-report items intended for measuring five areas regarding the health-related quality of life, namely physical and psychological wellbeing, autonomy and parent relations, peers and friends’ support, and school environment, respectively [32,33,34]. The third version, KIDSCREEN-10, is a unidimensional scale, containing 10 items regarding quality of life, representing one global score for the areas of the KIDSCREEN-27 version [35,36]. Normally, all three versions of KIDSCREEN address individuals aged 8–18 years and, aside from the self-report version, also have a proxy version for parents. In the present study, we chose to apply the KIDSCREEN-27 questionnaire on individuals aged 6, because we consider that 6-year-old children could respond to a health-related quality of life questionnaire and their answers could better explain the variations regarding the five areas of quality of life. We point out that this study belongs to a larger project, including, in addition to the KIDSCREEN questionnaire, other tests and measurements, which can provide important information about the level of physical development, basic motor skills or quality of life of children in Central European countries—Hungary, Romania and Slovakia [37]. Following the analysis of the obtained data, the results can represent useful information for the decision-making bodies regarding the structure of the educational program and the role of the physical activities for the harmonious development of the child [38].

Consequently, the purposes of the present research study are: (i) the validation of the KIDSCREEN-27 questionnaire for use with 6-year-old children, much younger than the children for whom this questionnaire is intended (children aged 8–18), and (ii) comparison regarding the quality of children’s lives, living in a provincial city in Romania (Arad) versus the quality of children’s lives, living in the second most developed city in Romania (Cluj-Napoca). Most likely, the differences obtained derive from the way of life in each city and from the opportunities that each city offers (jobs, institutions concerned with children’s activities, gyms, etc.). Following this study, we believe that we will provide an overview of the quality of life of Romanian 6-year-old children. As far as we know, this is the first research study in which the KIDSCREEN-27 questionnaire is applied to Romanian 6-year-old children in order to assess their quality of life.

## 2. Materials and Methods

### 2.1. Participants Selection and Design

In order to analyze the objectives of the cross-sectional study, the targeted participants consisted of the preparatory class children from schools of two cities in the west of the country, the second most developed city, Cluj-Napoca, and a provincial city, Arad. The present study was carried out within the international project EFOP-5.2.2-17-2017-00035—Complex comparative analysis of the interactions between regular health promotion physical activity, quality of life, motor abilities and body composition among students living in the Carpathian Basin within the framework of a Hungarian–Romanian–Slovak cooperation project. Some of the data obtained in the international project—more precisely the data obtained from the application of the KIDSCREEN-27 questionnaire to the children from the preparatory class—are statistically analyzed in this cross-sectional study. Data from the present study were gathered between November 2018 and October 2019. The cross-sectional study was divided in 3 phases (three assessment time points) and children in preparatory classes of schools from both cities were invited to participate in the study. A total of 342 preparatory class children, from both cities, alongside one or both parents expressed the desire to participate. The questionnaire was applied during classes and was anonymous. Both parents and children were given simple explanations of the questions in the questionnaire to confirm that they understood both the purpose of the research and their participation rights (data confidentiality, the right of withdrawal at any time). One parent of each involved child was asked to complete a questionnaire assessing the quality of life related to the child’s health. Children with medical problems were excluded from the study (n = 30). Twelve children aged above 7 years old were excluded. In the end, a total of 44 children from the initial group of participants, after exclusions, withdrew from the study during the 3 phases of the research (Figure 1). A total of 256 children participated in the study. The age of the children was between 73 and 81 months (mean 77.1, median 78 months and standard deviation 2.8).

### 2.2. Ethics Statements

Due to the fact that the present study involved human participants, the study was approved by the Ethics Committee of the Faculty of Physical Education and Sport, Aurel Vlaicu University of Arad (protocol code 252/24 May 2018), and was conducted in accordance with the Declaration of Helsinki. Moreover, all the children’s parents or legal guardians provided written informed consent before the study was conducted.

### 2.3. Instrument—KIDSCREEN-27

Regarding the quality life assessment of 6-year-old children, who have passed from the preschool stage to the school stage in a preparatory class, an assessment tool based on the KIDSCREEN-27 questionnaire was used [39]. In many European countries, the KIDSCREEN questionnaire has become the most suitable standardized instrument to assess the quality of children’s lives, with or without medical problems. It is worth mentioning that in Romania, only the KIDSCREEN-27 questionnaire is available in the mother tongue, this also being the reason why this questionnaire was chosen [40].

The KIDSCREEN-27 questionnaire is based on five health-related areas of investigation in which the physical (five items) and psychological wellbeing of the child (seven items), their autonomy and parent relations (seven items), peers and friends’ support (four items) as well as school environment (four items), with a total of 27 items, are evaluated. Each item has five response options depending on the intensity and frequency (not at all/never, slightly/seldom, moderately/quite often, very/very often and extremely/always). The response options of each item are coded on a 1- to 5-point scale. The questionnaire was administrated for the first time in November 2018, during classes, in the presence of the class teacher and two volunteer students from the Faculty of Physical Education and Sports, Aurel Vlaicu University of Arad and Babeș-Bolyai University of Cluj-Napoca, followed by another two assessment time points in May 2019 and October 2019. The re-test period for each evaluation phase was two weeks. It was chosen to apply the KIDSCREEN-27 questionnaire for three assessment time points, at a difference of five months between each evaluation phase, for two reasons: (i) we wanted to see if the thinking about the quality of life changes with children’s age, and (ii) we chose this long period between evaluation phases precisely so that the children could improve their quality of life.

Parents were asked to participate separately, so as not to influence their children’s responses. In order to get the answers as close as possible to the truth, the children were encouraged to be honest, as their name was not on the questionnaire. In the present study, the Romanian language self-report of the KIDSCREEN-27 was used [40].

Parents completed a questionnaire regarding their child’s date of birth, if the child does any sport and what kind of sport, how much training they participated in per week, the duration of the training as well as some data about the medical history of the child (chronic illness, what kind of illness, the lack or not of the motor development phase), which are not shown in the present analysis. In addition, they were asked about what kind of sport they practice, if appropriate, how much training they participated in per week and the duration of the training.

### 2.4. Statistical Analysis

The statistical analysis was performed using SPSS v. 27.0.0.0 64-bit edition statistical software (IBM SPSS Inc., Chicago, IL, USA) and Excel v. 1808 from MS Office Professional Plus 2019 (Microsoft, Redmond, WA, USA). Data are presented as categorical variables and frequency distributions. The distribution of variables was tested with the Kolmogorov–Smirnov test and the variables with normal distributions were presented as the mean value and standard deviation.

The differences between Arad and Cluj-Napoca children regarding health-related quality of life were determined using two-way ANOVA analysis followed by a Bonferroni post-test. In addition, for a better and more significant interpretation of the results, the false discovery rate (FDR) was used to increase the power of the test. Pearson correlation coefficients between the KIDSCREEN-27 areas of investigations were calculated. When correlations between comparable areas of investigations exceeded theoretical correlations between different areas of investigations, the convergent validity was considered to be demonstrated. The correlation coefficient intervals between 0.1 and 0.3, 0.3 and 0.5, and 0.5 or more were considered as low, moderate and large, respectively [41]. For the construct validity, it was started from the hypothesis that there are differences between healthy Arad and Cluj-Napoca children witha socioeconomic status from medium to high. Additionally, another test was done to compare the results of this research, namely the effect sizes calculated using Cohen’s d, computed as (M1−M2)/Sp, where M1 and M2 are the sample means for two groups and Sp denotes the pooled estimated population standard deviation. The effect sizes of 0.20–0.50, 0.50–0.80 and >0.80 were considered as weak, moderate and high, respectively [42].

The known-group validity of KIDSCREEN-27 was assessed by comparing the discrepancy of the health-related quality of life level between the children from both cities, at all three assessment time points. Eta square (η^2^) was computed and the following size magnitudes were considered: η^2^ = 0.01-small, 0.06-moderate and 0.14-large [31].

For the estimation of reproducibility, the intraclass correlation coefficient (ICC) was determined, by analyzing each dimension scale. ICC was calculated with the following equation:(1)$$ICC=\frac{\left[\left(mean-square\;between\;areas\right)-\left(mean-square\;within\;areas\right)\right]}{\left[\left(mean-square\;between\;areas\right)+\left(mean-square\;within\;areas\right)\right]}$$

An ICC of 0.6 or higher was considered acceptable due to the test–retest interval of 2 weeks.

To assess the internal consistency (IC), the Cronbach’s alpha coefficient was determined for each dimension.

## 3. Results

Before handing out the children’s questionnaire so that it could be completed, the two volunteer students had a group discussion with the children involved in the analysis about how they understand the concepts of health, wellbeing, autonomy and peer acceptance. All the children expressed their opinion related to each item on the questionnaire and the results obtained reflect their experiences and lifestyle until then. The KIDSCREEN-27 questionnaire was applied in three phases, with a re-test period for each evaluation phase of two weeks, on 128 healthy children of 6 years of age from Arad as well as from Cluj-Napoca. We chose to study 6-year-old children because they do not yet have the notion of school, thus facing an important challenge related to the transition from kindergarten to primary school, so they can be quite honest in their answers. At the beginning of this new stage of life, they still do not have the notion of competition, not being influenced by their parents. They are used to playing more, without a rigorous program imposed by the school. Through play and socializing, children learn to communicate with other children, to empathize, to express their emotions and feelings, to cooperate; in short, younger children are more satisfied with their lives. In this regard, we wanted to investigate whether their perception of quality of life changes with time. Most of the time, the notion of the new can create various problems. Older children are often overwhelmed by the volume of lessons taught and the amount of homework they receive, as well as the extracurricular activities they are often subjected to by their parents without their consent. For children older than 6 years, the psychological stress may be greater, and a poor diet or insufficient sleep may be more common. All this combined can lead to a decrease in quality of life and the appearance of various problems. Moreover, older children often face bullying. This phenomenon can lead to lower levels of emotional health, lower academic performance, difficulties in learning or adapting to school and even psycho-emotional disorders (depression and anxiety). FDA guidelines recommend that validation testing for children be performed in a fairly small age group, with the determination of the lower age limit at which children can provide valid and reliable answers [43]. In this context, Varni and co-workers [13] used data from the PedsQL™ Database^SM^ to test the hypothesis that children as young as 5 years of age can reliably and validly report their health-related quality of life. The results obtained by the authors showed that even 5-year-olds can reliably and validly self-report health-related quality of life when they have the opportunity to do so with an age-appropriate tool.

### 3.1. Differences between Arad and Cluj-Napoca Children Regarding the Health-Related Quality of Life

#### 3.1.1. Physical Activities and Health—Assessing the Child’s Physical Wellbeing

This area of investigation explores the degree of the 6-year-old child’s physical activity, their energy and their desire for practice sport. In addition, following this area of investigation, the measure in which a child feels unwell or complains of poor health can be assessed. The degree of physical activity refers to the child ability to be active at home and in school (to play or to do other physically demanding activities after school—swimming, football, biking, volleyball, handball, etc.).

Differences in the KIDSCREEN-27 scores by physical wellbeing, regarding the Arad vs. Cluj-Napoca children, are shown in Table 1.

According to Table 1, the lowest values were found in the case of item 1 and 3, both with an upward trend, while the highest values were recorded in the case of item 5, which also presents an upward trend in phase 3.

In the case of Arad children, statistically significant and moderate correlations were found between the 3rd and the 4th items (r = 0.51, *p* < 0.01) and between the 4th and the 5th items (r = 0.41, *p* < 0.01) at the 1st phase of evaluation. Statistically moderate correlations were also found between the 2nd and the 3rd items (r = 0.45, *p* < 0.01), between the 2nd and the 4th items (r = 0.46, *p* < 0.01) and between the 3rd and the 4th items (r = 0.48, *p* < 0.01) at the 2nd phase of evaluation, while the last phase of evaluation revealed a correlation between the 2nd and the 4th items (r = 0.44, *p* < 0.01).

In the case of Cluj-Napoca children, two statistically moderate correlations were found between the 2nd and the 3rd questions and between the 3rd and the 4th questions (both with r = 0.41, *p* < 0.01), respectively, in the 1st phase of evaluation; only weak correlations were observed in the 2nd evaluation (r < 0.3). Similar to the other group, the last evaluation revealed a moderate correlation between the 2nd and the 4th question (r = 0.41, *p* < 0.01).

#### 3.1.2. General Mood and How You Feel about Yourself—Psychological Wellbeing

The second area of investigation explores the life satisfaction of the child with their positive emotions, exactly as they are perceived by the child and experienced. Moreover, each answer of the child reflects how much a child experiences positive or negative emotions; what joy, happiness and cheerfulness mean to the; as well as how satisfied they are with the life they have lived up to that moment. Following this area of investigation, the presence or absence of some feelings (loneliness and/or sadness) can be evaluated.

Differences in the KIDSCREEN-27 scores by psychological wellbeing, regarding the Arad vs. Cluj-Napoca children, are shown in Table 2.

According to Table 2, the lowest values were found in the case of item 7, but it is worth mentioning that no mean value was below 4 (meaning the children’s answers were only “very often” and “always”); the highest values in this set were recorded in the case of items 3 and 4, which also present an upward trend, except for item 4 in the Arad group, where the mean value decreased continuously for the entire period of evaluation.

In the case of Arad children, one statistically significant and moderate correlation was found between the 5th and the 6th items (r = 0.52, *p* < 0.01) at the 1st phase of evaluation, then between the 1st and the 3rd items (r = 0.47, *p* < 0.01) and between the 1st and the 7th items (r = 0.50, *p* < 0.01) from the 2nd phase of evaluation, while the last phase of evaluation revealed just weak correlations (r < 0.40).

In the case of the other group of children (Cluj-Napoca), two statistically significant and moderate correlations were found between the 1st and the 2nd items (r = 0.42, *p* < 0.01) and between the 2nd and the 3rd items (r = 0.50, *p* < 0.01), respectively, at the 1st phase of evaluation; two statistically significant and moderate correlations were found between the 1st and the 3rd items (r = 0.43, *p* < 0.01) and between the 2nd and the 3rd items (r = 0.56, *p* < 0.01), respectively, at the 2nd phase of evaluation. The following correlations were found in the last phase of evaluation: between the 2nd and the 3rd items (r = 0.50, *p* < 0.01), between the 4th and the 5th items (r = 0.50, *p* < 0.01) and between the 5th and the 6th items (r = 0.51, *p* < 0.01).

#### 3.1.3. Family and Free Time—Autonomy and Parent Relations

This area of investigation explores both the relationship between parents as well as between the parents and child, and the child’s feelings about their parents. Also, it can reveal the atmosphere in the child’s home, if they are loved, accepted, understood and supported no matter what. Regarding autonomy, this investigation can bring to the fore the child’s independence and self-support or self-sufficiency. Particularly considered is the child’s ability to make decisions about their own life or about activities which they practice or want to practice day-to-day. This area of investigation refers to the child’s degree of autonomy, and how capable they are of creating a unique identity to manage their social and leisure time. Some items are about the child’s perception regarding the financial resources of the family.

Table 3 shows the differences between Arad and Cluj-Napoca children in the KIDSCREEN-27 scores related to items about family and free time.

According to Table 3, the lowest values were found in the case of items 3, 6 and 7, while the highest values were established in the case of item 4 where the evolution almost remained constant in Arad. Regarding the children’s answers for item 5, there was a downward trend in Arad, which is exactly the opposite of what was observed in Cluj-Napoca.

In the case of Arad children, the following correlation were found in the 1st phase of evaluation: a statistically significant correlation between the 1st and the 2nd items (r = 0.53, *p* < 0.01), a moderate correlation between the 2nd and the 7th items (r = 0.47, *p* < 0.01) and a statistically significant correlation between the 4th and the 5th items (r = 0.51, *p* < 0.01) and between the 6th and the 7th items (r = 0.61, *p* < 0.01). Moderate and significant correlations between the 1st and the 2nd items (r = 0.47, *p* < 0.01), between the 2nd and the 3rd items (r = 0.45, *p* < 0.01), between the 2nd and the 4th items (r = 0.52, *p* < 0.01) and between the 6th and the 7th items (r = 0.64, *p* < 0.01) were observed at the 2nd phase of evaluation, while the last phase of evaluation revealed just one significant correlation between the 6th and the 7th items (r = 0.51, *p* < 0.01).

For the Cluj-Napoca children, a statistically significant correlation between the 6th and the 7th items (r = 0.52, *p* < 0.01) was found at the 1st phase of evaluation; two statistically significant and moderate correlations were found between the 3th and the 6th items (r = 0.46, *p* < 0.01) and between the 6th and the 7th items (r = 0.61, *p* < 0.01), respectively, at the 2nd phase of evaluation. The last phase of evaluation contained only moderate correlations: between the 1st and the 2nd items (r = 0.47, *p* < 0.01), between the 3rd and the 5th items (r = 0.47, *p* < 0.01) and between the 6th and the 7th items (r = 0.46, *p* < 0.01).

#### 3.1.4. Peer Acceptance and Friends’ Support

This area of investigation explores first of all the child’s relationships with their friends or with other children of the same age, younger or older. Following this area of investigation, it can be predicted how often the child interacts with their friends, peers or children from their family (relatives) as well as in what manner the child feels accepted and supported by them. In addition, this dimension analyzes the child’s ability to make new and maintain old friendships. The emphasis is on the communication with other children, on the perceived support and in what manner the child feels part of a group and respected by peers and friends.

Table 4 show the differences between children of Arad and Cluj-Napoca in the KIDSCREEN-27 scores related to items about friends support.

According to Table 4, the lowest average values were found in the case of item 4 in the case of Arad children and item 1 in the case of Cluj-Napoca children, both with an upward trend, while the highest values were established in the case of item 2 in both cities with the same evolution in time.

In the case of Arad children, there were moderate correlations between the 1st and the 4th items (r = 0.42, *p* < 0.01) and the 3rd and the 4th items (r = 0.42, *p* < 0.01) at the 1st phase of evaluation. Moderate correlation between the 1st and the 2nd items (r = 0.45, *p* < 0.01) was noticed at the 2nd phase of evaluation, while the last phase of evaluation revealed just one moderate correlation between the 1st and the 2nd items (r = 0.48, *p* < 0.01).

In the case of Cluj-Napoca children, a correlation between the 2nd and the 3rd items (r = 0.49, *p* < 0.01) was found at the 1st phase of evaluation; two statistically significant and moderate correlations were found between the 1st and the 2nd items (r = 0.60, *p* < 0.01) and between the 3rd and the 4th items (r = 0.45, *p* < 0.01), respectively, at the 2nd phase of evaluation. The last phase of evaluation contained the following correlations: between the 1st and the 2nd items (r = 0.46, *p* < 0.01) and between the 3rd and the 4th items (r = 0.47, *p* < 0.01).

#### 3.1.5. School Environment—School and Learning

This area of investigation explores how the child perceives school, the feelings related to school as well as the satisfaction with going to school, how they consider if the school is an enjoyable place or not. This dimension also explores the cognitive capacity of the child, how much they like to learn, their capacity of concentration at school and their abilities and school performance. Moreover, the child’s perception about their class teacher is also brought to the surface; this dimension contains some items related to the child’s relationship with the class teacher.

Table 5 show the differences between children of Arad and Cluj-Napoca in the KIDSCREEN-27 scores related to items about school environment.

According to Table 5, the lowest average values were for item 3, but it is important to mention that they are above 4.20; the highest values were established in the case of item 2 regarding the Arad children. An interesting descending evolution was found between the results of the 1st item in the group of Cluj-Napoca children.

In the case of Arad children, there was a moderate correlation just between the 2nd and the 4th items (r = 0.41, *p* < 0.01) at the 1st phase of evaluation. Moderate correlations between the 2nd and the 4th items (r = 0.41, *p* < 0.01) and between the 3rd and the 4th items (r = 0.49, *p* < 0.01) were noticed at the 2nd phase of evaluation, while the last phase of evaluation revealed just one significant correlation between the 1st and the 2nd items (r = 0.56, *p* < 0.01).

For the group of Cluj-Napoca children, a moderate correlation between the 2nd and the 3rd items (r = 0.41, *p* < 0.01) was found at the 1st phase of evaluation; three statistically significant and moderate correlations were found between the 1st and the 2nd items (r = 0.53, *p* < 0.01), between the 1st and the 3rd items (r = 0.47, *p* < 0.01) and between the 3rd and the 4th items (r = 0.52, *p* < 0.01), respectively, at the 2nd phase of evaluation. The last phase of evaluation contained only one moderate correlation between the 1st and the 2nd items (r = 0.46, *p* < 0.01).

### 3.2. Reliability Analysis

The internal consistency was assessed using Cronbach’s alpha coefficient for each area of investigation at all three assessment time points. For the test–retest reliability, the participants completed the questionnaire twice at each phase of evaluation, at a 2-week interval, which was considered long enough to ensure that they would not remember their first responses, but short enough to avoid important modifications in the quality of their lives.

The Cronbach’s alpha coefficient for this questionnaire was 0.683, while the ICC for test–retest analysis, responsible for reproducibility, ranged from 0.519 (2nd area of investigation) to 0.655 (5th area of investigation) at the end of the study (after 1 year) (Table 6).

### 3.3. Validity

Convergent validity analysis is shown in Table 7. We started from the hypothesis that there are differences between healthy Arad and Cluj-Napoca children with a socioeconomic status from medium to high. The effect sizes calculated using Cohen’s d generally indicate a moderate to high level of correlation for the 3rd, 4th and the 5th areas of investigation and a low level of correlation for the first two areas of investigation.

The known-group validity was assessed by comparing the discrepancy of the health-related quality of life level between children of both cities regarding their socioeconomic statuses, academic performance and health statuses. The differences obtained are shown in Table 8. It can be seen that, regarding the socioeconomic status and academic performance, a moderate effect size was obtained compared with a low to moderate effect size on health status.

The false discovery rate (FDR), a method of conceptualizing the rate of type I errors in testing of the null hypothesis when multiple comparisons are performed, was previously described in the literature by Benjamini and Hochberg [44], and it was calculated using a code for SPSS. FDR is the ratio FP/(FP + TP), where FP is false positives, TP is true positives, while (FP + TP) is the total number of rejections of the null hypothesis [45]. Following the step-by-step procedure previously described by Cramer et al. [45], the obtained adjusted *p*-values regarding our study are still significant.

## 4. Discussion

The present study can be considered the first study to validate a Romanian language version of a health-related quality of life questionnaire applied to a population of 6-year-old preparatory school children living in two cities from Romania: Arad and Cluj-Napoca. Arad is the 12th largest city in Western Romania, with a population of 159,704 inhabitants [46]. Arad is considered the most important trans-European road and rail transportation junction point in western Romania, linking Western Europe to South-Eastern European and Middle Eastern countries. Cluj-Napoca is sited in the northwestern part of the country and is the fourth most populous city in Romania [47]. In 2011, Cluj-Napoca comprised 324,576 inhabitants living within the city limits, which made it the country’s second most populous city at the time, after the national capital Bucharest [46]. As a metropolitan area, Cluj-Napoca has a population of 411,379 people. Table 9 highlights a few differences between these two major cities of Romania, because it is important to mention that every data/parameter influences more or less the quality of life standards of citizens.

In the present study, the research focuses only on healthy children who are living with both parents and have a medium to high socioeconomic status. The KIDSCREEN-27 questionnaire was taken into consideration due to the fact that is the first instrument for children which was developed simultaneously and tested in several countries. In addition, in Romania, of the three versions of the questionnaire it is the only one available in the mother tongue. The vast majority of research studies focus on either children with various health problems or children who are orphaned or come from divided families [52,53,54,55,56,57,58]. There are few published studies investigating the health-related quality of life, the needs and desires, and the physical and mental wellbeing of children without medical problems who come from families with both parents and with a socioeconomic status from medium to high [11,59,60,61]. As far as we know, none of these studies evaluated 6-year-old healthy children, especially Romanian children. As for the Romanian children with medical problems, Urzeală and co-workers assessed the quality of life in type 1 diabetes mellitus (T1DM) children who attended an early interdisciplinary healthcare intervention [52]. The authors started from the hypothesis that the children’s quality of life would increase with the T1DM children’s participation in leisure sports. As a research method, the KIDSCREEN-27 questionnaire was applied to 100 T1DM children aged between 7 and 17 years. All the evaluated children reached an increased level of all areas of investigations (physical and psychological wellbeing, autonomy and parent relations, peers and friends’ support, and school environment). Regarding the physical wellbeing, a statistically significant differences (*p* < 0.05) was obtained between the children who practiced leisure sports and children who only participated in physical education classes [52].

In the context of an acute lack of information regarding the health-related quality of life of Romanian children, the present study aims to bring a minimal contribution to the health-related quality of life in terms of physical and mental wellbeing, self-esteem and the needs and desires of preparatory school children living in two different cities of Romania. The KIDSCREEN-27 questionnaire includes five areas of investigation covering physical, psychological and social aspects of health-related quality of life. It is important to investigate the general wellbeing of a healthy child, as well as the psychological wellbeing, alongside general mood, emotions and peer acceptance. Moreover, autonomy, relationship with parents and school and learning subscales were included. One of the objectives of this paper was to understand the 6-year-old children’s self-perception of health-related quality of life, due to the fact that the investigated children were at an important stage in their lives, the beginning of school.

The outcomes of this study confirm that the Romanian version of the KIDSCREEN-27 shows an acceptable degree of validity and reliability, as well as psychometric properties for 6-year-old children in preparatory school. Two-way ANOVA analysis followed by a Bonferroni post-test was conducted to explore the differences between items for each area of investigation, and it statistically significant differences were found. According to Bortz and Doring [41], and taking into consideration the parameters of the present study (healthy children coming from families with both parents and with a medium to high socioeconomic status), a Pearson correlation coefficient below 0.4 was considered low, between 0.4 and 0.5 was considered moderate and above 0.5 was considered statistically significant.

Overall, the children from both cities (Arad and Cluj-Napoca) scored higher for all the areas of investigation of the KIDSCREEN-27 (Table 1, Table 2, Table 3, Table 4 and Table 5). For the first area of investigation (physical wellbeing—physical activities and health), the differences obtained were moderately significant (r = 0.41–0.48) between items 2–4 for the children living in both cities and statistically significant (r = 0.51) only for the children of Arad, between the 3rd and the 4th items at the first phase of evaluation. This item refers to the children’s response regarding the abilities for physical activities (running, climbing, biking, etc.). A weak correlation was obtained at the second phase of evaluation for the children of Cluj-Napoca (r < 0.3) between the 2nd and 3rd as well as the 3rd and 4th items. As we expected, children of Cluj-Napoca, who have a higher standard of living and participate in at least two physical activities per week, showed a poor correlation between the considered items, which means that the health-related quality of life of these children is quite good compared with children of Arad, whose answers showed that, overall, their health-related quality of life level measured by the KIDSCREEN-27 questionnaire demonstrated significant differences among socioeconomic statuses/health statuses related to physical activities. On the other hand, it is worth mentioning that the importance of physical activities is related to the age and the body mass index of children, as well as having an impact on their mood; unfortunately, more and more children nowadays spend a lot of time in front of different electronic devices despite their need for outdoor activities. In our studied group, the age of these children was perfect to determine their chance of developing obesity and psychosocial problems and to influence their behavior towards outdoor movement [16].

Regarding the second area of investigation, in the case of Arad children, a significant correlation (r = 0.52) was found when the children were asked if they felt lonely or if they were in a bad mood such that they did not feel like doing anything. According to Ravens-Sieberer and co-workers [35], these two items report information about the mental health of the child (emotions, depressive and/or stressful moods and feelings). About the same result (r = 0.50) was obtained between items 1 (*Has your life been enjoyable?*) and 7 (*Have you been happy with the way you are?*). Apparently, this bad mood, along with the feeling of loneliness and sometimes unhappiness, can affect the child’s health, and from here to their illness there is only one small step left. These results indicate that worse psychological wellbeing is related to a worse health-related quality of life. However, it is worth mentioning that a weak correlation (r < 0.4) was obtained regarding the items described above at the last phase of evaluation. This improvement is due to the fact that, after each evaluation phase, the specialists involved in the study spoke in particular with each of the children’s parents whose answers led to statistically significant correlations. The specialists explained in detail what a significant impact physical activity has on the child’s mental health and how important it is that any sport practiced, even a simple walk in the open air, brings happiness and positive feelings, leading to the increase of the child’s quality of life.

Ferreira and co-workers [62] tested the reliability and validity of the pediatric quality of life inventory (PedsQL) for Portuguese children aged 5–7 and 8–12 years, including healthy children, children with type I diabetes and children who suffer with spina bifida. They found that the psychosocial dimension (emotional, social and school functioning) for the 5–7-year-old children version showed a Pearson correlation score below 0.70. In addition, concordance values between children’s and parents’ perceptions ranged between 0.36 and 0.78.

The most significant statistical correlations were obtained following the evaluation of the third area of investigation—the child’s autonomy and the relationship with their parents—both for the Arad children as well as for the Cluj-Napoca children. The Pearson correlation coefficients were statistically significant (r = 0.53, group of Arad) between items 1 (*Have you had enough time for yourself?*) and 2 (*Have you been able to do things that you want to do in your free time?*) at the first phase of evaluation, becoming moderate correlations at the next phases of the evaluation. The results obtained suggest that the relationship between children and their parents improved during one year of evaluation, meaning that, the latter gave the children more autonomy and the freedom to decide how to spend their free time, which led to an increase in their self-esteem. The parents were also open-minded: they understood the advice of the specialists involved in the study regarding the child’s need to be independent, to decide for themselves what extracurricular activities they are passionate about without being constrained by them. The fact that a child, even of 6 years of age, is not allowed to manage their free and social time, of course after a previous guidance, can only lead to the deterioration of the relationship between them and their parents. The child may feel unaccepted, misunderstood or unloved, all of which lead to a deterioration in their quality of life. Another statistically significant correlation coefficients (r = 0.61, group of Arad; r = 0.52, group of Cluj-Napoca) were obtained when the children were asked if they had enough money to do the same things as their friends (item 6) and if they had enough money for their expenses (item 7). In the case of Arad children, the results regarding both items increased slightly in the second phase of evaluation (r = 0.64) to then decrease quite a bit in the third evaluation phase (r = 0.51). Almost the same happened in the case of children from Cluj-Napoca, with the mention that, in the second phase of evaluation, the Pearson correlation coefficient increased more (r = 0.61) and then decreased dramatically in the third phase of evaluation (r = 0.46). These results do not fit with the socioeconomic status of the population of Cluj-Napoca. It is very possible that the discussions between the parents regarding the financial resources of the family may be misunderstood by the child, hence the increase of the Pearson correlation coefficient in the second stage of evaluation. However, in parents’ perception, these results can be strongly correlated with the age of the children; it goes without saying that a child in a preparatory school does not have their own expenses at school, considering that they are not allowed to leave the school premises. Our results are in accordance with those obtained by da Silveira and co-workers, which verified the reliability, discriminatory power and construct validity of the KIDSCREEN-27 questionnaire in Brazilian adolescents (pilot sample 13.7 ± 1.0 years, baseline sample 13.1 ± 1.1 years) [61]. They found that the Pearson correlation coefficient were greater than 0.40, except for item 7 (*Have you had enough money for your expenses?*) (r = 0.36). Regarding item 6, the Pearson correlation was 0.43 (*Have you had enough money to do the same things as your friends?*), slightly higher than our values obtained in both cities.

Regarding the fourth area of investigation—peers and friends’ support—the Pearson correlation coefficients were statistically moderate for the Arad children in all the items evaluated (r = 0.42 ÷ 0.48). In the case of Cluj-Napoca children, at the first phase of evaluation a statistically significant correlation (r = 0.60) was obtained between the first two items (*Have you spent time with your friends?*/*Have you had fun with your friends?*), which decreased in the last phase of evaluation (r = 0.46). The results obtained after evaluating this dimension fit very well with the results of Cluj-Napoca children obtained after evaluating the first dimension (physical wellbeing). Correlating these results with the results obtained from Cluj-Napoca children in items 3 and 4 from the first area of investigation, the lack of time spent with friends and the desire to have fun or do things with them are understandable. Obviously, the specialists involved in the study explained to both parents and children how important it is to socialize with peers and how much self-confidence is developed when the child feels accepted, understood and supported by other individuals who are not part of their family. Following the discussions, the parents were advised to find a way in which the child could allocate time to friends without neglecting the physical activities in which they participate. Our results (r = 0.46–at the last phase of evaluation) are much smaller than those obtained by da Silveira [61], who obtained a Pearson correlation of 0.93 for item 2 (*Have you had fun with your friends?*)

The last area investigated—learning and school environment—revealed statistically moderate coefficients for the Arad children (r = 0.41 ÷ 0.49) and only one statistically significant correlation (r = 0.56) in the last phase of evaluation, between the first two items (*Have you been happy at school?*/*Have you got on well at school?*). Almost the same result was obtained in the case of children from Cluj-Napoca (r = 0.53), but in the second phase of evaluation, following by a moderate correlation (r = 0.46) in the third phase of evaluation. In addition, also in the second phase of evaluation, the results of the children from Cluj-Napoca showed a statistically significant correlation between the 3rd and 4th items (*Have you been able to pay attention?*/*Have you got along well with your teacher?*). This is explained by the fact that children who are often subjected to many physical activities, either by their own desire or that of their parents, tend to be more tired in the school environment, thus leading to a decrease in their cognitive capacity. Moreover, decreased ability to concentrate at school, skills and school performance can change the child’s perception about school and also reduce the satisfaction of attending school, thus changing their labeling of the school as a desirable place. Similar results regarding the learning and school environment area of investigation were obtained by Ravens-Sieberer and co-workers [63], who measured quality of life and well-being in children aged 11, 13 and 15 years, attending regular schools, by applying the KIDSCREEN-27 questionnaire (T value = 0.501).

Regarding reproducibility, the ICC ranged from 0.584 (learning and school environment) to 0.758 (autonomy and parent relations) in the first phase of evaluation; from 0.542 (psychological wellbeing) to 0.721 (learning and school environment) in the second phase of evaluation and from 0.511 (peers and friends’ support) to 0.655 (learning and school environment) in the third phase of evaluation (Table 6).

The measure of internal consistency (IC) assessed by Cronbach’s alpha ranged from 0.598 (learning and school environment) to 0.776 (autonomy and parent relations) in the first phase of evaluation; from 0.585 (psychological wellbeing) to 0.766 (learning and school environment) in the second phase of evaluation; and from 0.554 (peers and friends’ support) to 0.661 (learning and school environment) in the third phase of evaluation (Table 6). Similar results were obtained by Shannon and co-workers [59], who tested the psychometric properties of KIDSCREEN-27 with Irish children 8–9 years old with low socioeconomic status. The authors reported that the Cronbach’s alpha values ranged from 0.56 to 0.74. Moreover, thy found that the moods and emotions were the only subscales below the recommended value of 0.6, as in our case (IC = 0.585). Pardo-Guijaro and co-workers [54] reported a Cronbach’s alpha below 0.75 when they tested the validity and reliability of the Spanish sign language version of the KIDSCREEN-27 health-related quality of life questionnaire on deaf children and adolescents (8 to 18 years old).

Both ICC and IC showed a linear decreasing trend after each evaluation phase in three of the five areas of investigation, namely psychological wellbeing, autonomy and parent relations, and peers and friends’ support. These three dimensions have improved over 1 year of evaluation, which represents a success regarding the health-related quality of life, especially for 6-year-old children. Due to the 2-week interval test–retest, an ICC of 0.6 or higher was considered acceptable regarding the reproducibility of the data. The reproducibility of the present study found lower ICC values than other studies [32,34,60,61,64,65,66]. The differences are mainly due to the size of the study, the age range of the children (6 years in the present study versus over 8 years in the other studies), the retest period (2 weeks in our study versus 1 week in the other studies) and the questionnaire application procedure (questionnaire completion during classes versus face-to-face interview or telephonic interview).

Cohen’s d coefficient is obtained by dividing the differences of the adjusted means at the overall standard deviation [42]. Convergent validity was indicated by the association of children of Arad and Cluj-Napoca between the KIDSCREEN-27 areas of investigation. Correlations were generally moderate to high for those areas where higher Pearson correlations were obtained (autonomy and parent relations, peers and friends’ support, and learning and school environment). An exception was the second area of investigation in which the most statistically significant Pearson correlations were obtained, but the effect sizes calculated using Cohen’s d indicated low levels of correlation.

Differences in the scores of this questionnaire by socioeconomic status, health status and academic performance have been calculated based on the techniques that were described by Gong and co-workers [67]. A moderate effect size among participants with different socioeconomic statuses (η^2^ = 0.061, F-value = 26.76, *p* < 0.001) and with different academic performance (η^2^ = 0.067, F-value = 29.35, *p* < 0.01) and a low to moderate effect size on health status (η^2^ = 0.034, F-value = 18.59, *p* < 0.001) were obtained. Apparently, the quality of life from the perspective of 6-year-old preparatory school children is influenced more by socioeconomic status and academic performance than health status.

Overall, the KIDSCREEN-27 questionnaire showed good results in terms of convergent validity and known-group validity. The hypothesis from which we departed was verified and it was shown that there are some differences between healthy Arad and Cluj-Napoca children with a socioeconomic status from medium to high, especially in the last three dimensions (autonomy and parent relations; peers and friends’ support; and school environment), in which Cohen’s d ranged from 0.50 to 0.73 (Table 7). According to our results and taking into consideration that the health-related quality of life instruments are meant to divide children and adolescents in different socioeconomic categories, it can be affirmed that the differences between Arad and Cluj-Napoca children came from the salary income being higher in Cluj-Napoca than Arad.

The present study has the following limitations. First, there was a small sample size, particularly due to the fact that we took into consideration only healthy children coming from complete families with a socioeconomic status from medium to high. Given that the sample consisted of minor children who had to be accompanied by parents, the target population is often not open to this type of assessment, mainly in underdeveloped cities. In Cluj-Napoca, for example, we had an extraordinarily high participation rate of the target population, but the small number of participants in Arad led to a reduction of the data obtained in Cluj-Napoca. Second, there was the children’s inability to read the questions in the questionnaire on their own. This aspect can be a disadvantage for the study due to the fact that the children were in a preparatory class and, not knowing how to read, the teacher in the class had read the question to them, even twice, and the members of the study team had to explain to them, with the help of the board, how to tick one of the answers depending on their choice. Third, another limitation could be the fact that the gender of the children (male, female) from both cities was not taken into account. It would certainly have been interesting to analyze children’s responses by gender. Likewise, for a clearer interpretation of the results, it would have been good to divide the children into two social classes, the middle class and the class of children with above average social status. However, following discussions with parents, we proceeded with the premise that middle-class families also have enough financial resources to support at least one physical activity per week for the child.

Last but not least, another limitation is related to the timing of our study—the start of the school year vs. later in the year; the observed children face an important challenge related to the transition from kindergarten to primary education. Thus, scores of the items from this questionnaire related to their general mood (the second area of investigation) and to school environment (last area of investigation) may be prejudiced.

## 5. Conclusions

KIDSCREEN-27 still remains a powerful instrument for generic and cross-cultural health-related quality of life measurements. In this study, it was demonstrated that the KIDSCREEN-27 questionnaire can be applied to 6-year-old children, with reliable and reproducible results. In this assessment process it achieved good levels of reliability assessed by test–retest reproducibility and internal consistency (assessed by Cronbach’s alpha) as well as good validity levels. We have shown in this study that, without a doubt, the practice of at least one physical activity per week leads to an increase in the quality of life of children. However, the child’s physical overload and the pressure of learning given by the beginning of a new stage in their life can lead to the development of negative feelings (sadness, loneliness, depression, etc.) towards the life they live, school, extracurricular activities and even towards their own parents. A balance must be found between the wishes and needs of the child and the wishes of the parents.

In summary, the KIDSCREEN-27 questionnaire can be used as a reliable cross-cultural valid self-administered instrument for assessing 6-year-old children’s health-related quality of life, as reflected by its conceptual and methodological strengths.

## Figures and Tables

**Figure 1 healthcare-10-01198-f001:**
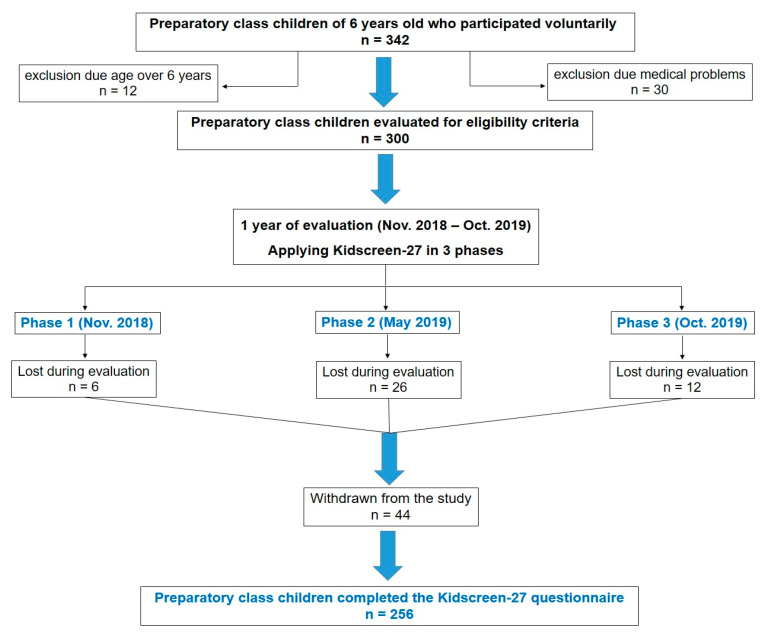
Flowchart of the study participants from both cities.

**Table 1 healthcare-10-01198-t001:** Evolution of mean values for the Arad and Cluj-Napoca children regarding the answers of items from the first area of investigation—physical wellbeing.

Item	Mean Values ± SD					
	Arad			Cluj-Napoca		
	Phase 1—Nov. 2018	Phase 2—May 2019	Phase 3—Oct. 2019	Phase 1—Nov. 2018	Phase 2—May 2019	Phase 3—Oct. 2019
1	3.77 ± 1.06	4.23 ± 1.05	3.97 ± 1.03	3.33 ± 1.26	4.03 ± 0.99	4.17 ± 0.96
2	4.11 ± 0.96	4.41 ± 0.99	4.38 ± 0.83	4.08 ± 1.11	4.37 ± 0.81	4.38 ± 0.85
3	3.95 ± 1.09	4.29 ± 1.07	4.05 ± 1.11	3.77 ± 1.25	4.26 ± 1.11	4.23 ± 1.08
4	4.12 ± 1.01	4.35 ± 1.08	4.30 ± 1.02	4.04 ± 1.15	4.29 ± 1.07	4.34 ± 1.07
5	4.24 ± 1.04	4.47 ± 0.79	4.45 ± 0.87	4.15 ± 1.09	4.50 ± 0.92	4.31 ± 0.99

Item 1—In general, how would you say your health is? Item 2—Thinking about the last week, have you felt fit and well? Item 3—Thinking about the last week, have you been physically active—running, climbing, biking? Item 4—Thinking about the last week, have you been able to run well? Item 5—Thinking about the last week, have you felt full of energy?

**Table 2 healthcare-10-01198-t002:** Evolution of mean values for the Arad and Cluj-Napoca children regarding the answers of items from the second area of investigation—psychological wellbeing.

Item	Mean Values ± SD					
	Arad			Cluj-Napoca		
	Phase 1—Nov. 2018	Phase 2—May 2019	Phase 3—Oct. 2019	Phase 1—Nov. 2018	Phase 2—May 2019	Phase 3—Oct. 2019
1	4.30 ± 0.87	4.68 ± 0.56	4.48 ± 0.73	4.27 ± 0.89	4.49 ± 0.80	4.34 ± 0.89
2	4.24 ± 0.87	4.32 ± 1.00	4.17 ± 1.02	4.10 ± 1.09	4.48 ± 0.78	4.22 ± 1.00
3	4.44 ± 0.82	4.32 ± 1.09	4.54 ± 0.76	4.20 ± 1.04	4.62 ± 0.67	4.46 ± 0.86
4	4.48 ± 0.88	4.41 ± 0.93	4.24 ± 0.99	4.16 ± 0.85	4.28 ± 0.85	4.29 ± 0.87
5	4.09 ± 1.23	4.57 ± 0.66	4.27 ± 0.99	4.41 ± 0.79	4.52 ± 0.60	4.45 ± 0.84
6	4.15 ± 1.36	4.57 ± 0.74	4.41 ± 0.98	4.20 ± 1.09	4.53 ± 0.74	4.41 ± 0.96
7	4.00 ± 1.24	4.41 ± 0.89	4.23 ± 1.08	4.08 ± 1.17	4.16 ± 1.20	4.30 ± 0.94

Item 1—Thinking about last the week, has your life been enjoyable? Item 2—Thinking about the last week, have you been in a good mood? Item 3—Thinking about the last week, have you had fun? Item 4—Thinking about the last week, have you felt sad? Item 5—Thinking about the last week, have you felt so bad that you didn’t want to do anything? Item 6—Thinking about the last week, have you felt lonely? Item 7—Thinking about the last week, have you been happy with the way you are?

**Table 3 healthcare-10-01198-t003:** Evolution of mean values for the Arad and Cluj-Napoca children regarding the answers of items from the third area of investigation—autonomy and parent’s relationship.

Item	Mean Values ± SD					
	Arad			Cluj-Napoca		
	Phase 1—Nov. 2018	Phase 2—May 2019	Phase 3—Oct. 2019	Phase 1—Nov. 2018	Phase 2—May 2019	Phase 3—Oct. 2019
1	4.16 ± 1.09	4.42 ± 0.93	4.13 ± 1.05	3.97 ± 1.18	4.00 ± 1.12	4.03 ± 1.00
2	4.01 ± 1.18	4.27 ± 1.17	4.06 ± 1.14	3.89 ± 1.21	4.12 ± 1.07	3.95 ± 1.14
3	3.67 ± 1.34	4.03 ± 1.16	3.91 ± 1.22	3.57 ± 1.40	3.59 ± 1.26	3.74 ± 1.05
4	4.09 ± 1.22	4.33 ± 0.99	4.13 ± 1.26	4.05 ± 1.18	4.23 ± 1.01	4.31 ± 0.86
5	4.24 ± 1.22	4.28 ± 1.00	4.13 ± 1.20	3.96 ± 1.29	4.20 ± 1.05	4.11 ± 1.05
6	3.68 ± 1.30	3.54 ± 1.44	3.58 ± 1.43	3.17 ± 1.49	3.34 ± 1.24	3.30 ± 1.34
7	3.64 ± 1.40	3.71 ± 1.46	3.90 ± 1.30	3.25 ± 1.55	3.50 ± 1.29	3.63 ± 1.37

Item 1—Thinking about last the week, have you had enough time for yourself? Item 2—Thinking about the last week, have you been able to do the things that you want to do in your free time? Item 3—Thinking about the last week, have your parent(s) had enough time for you? Item 4—Thinking about the last week, have your parent(s) treated you fairly? Item 5—Thinking about the last week, have you been able talk to your parent(s) when you wanted to? Item 6—Thinking about the last week, have you had enough money to do the same things as your friends? Item 7—Thinking about the last week, have you had enough money for your expenses?

**Table 4 healthcare-10-01198-t004:** Evolution of mean values for the Arad and Cluj-Napoca children regarding the answers of items from the fourth area of investigation—peers and friends’ support.

Item	Mean Values ± SD					
	Arad			Cluj-Napoca		
	Phase 1—Nov. 2018	Phase 2—May 2019	Phase 3—Oct. 2019	Phase 1—Nov. 2018	Phase 2—May 2019	Phase 3—Oct. 2019
1	4.14 ± 1.09	4.33 ± 1.04	4.38 ± 0.89	3.92 ± 1.30	4.10 ± 1.13	4.27 ± 1.13
2	4.36 ± 1.03	4.49 ± 0.97	4.70 ± 0.61	4.31 ± 1.07	4.48 ± 0.87	4.54 ± 0.85
3	4.17 ± 1.01	4.20 ± 1.11	4.40 ± 0.95	4.05 ± 1.25	4.37 ± 1.00	4.34 ± 0.91
4	3.96 ± 1.15	4.21 ± 1.16	4.21 ± 1.08	4.05 ± 1.22	4.17 ± 1.12	4.08 ± 1.25

Item 1—Thinking about last the week, have you spent time with your friends? Item 2—Thinking about the last week, have you had fun with your friends? Item 3—Thinking about the last week, have you and your friends helped each other? Item 4—Thinking about the last week, have you been able to rely on your friends?

**Table 5 healthcare-10-01198-t005:** Evolution of mean values for the Arad and Cluj-Napoca children regarding the answers of items from the fifth area of investigation—school environment.

Item	Mean Values ± SD					
	Arad			Cluj-Napoca		
	Phase 1—Nov. 2018	Phase 2—May 2019	Phase 3—Oct. 2019	Phase 1—Nov. 2018	Phase 2—May 2019	Phase 3—Oct. 2019
1	4.41 ± 0.75	4.53 ± 0.90	4.50 ± 0.79	4.48 ± 0.83	4.48 ± 0.74	4.33 ± 0.92
2	4.45 ± 0.80	4.41 ± 0.88	4.63 ± 0.66	4.14 ± 1.07	4.45 ± 0.76	4.40 ± 0.68
3	4.20 ± 1.09	4.42 ± 0.94	4.41 ± 0.90	4.20 ± 1.11	4.45 ± 0.82	4.46 ± 0.80
4	4.41 ± 0.89	4.52 ± 0.93	4.53 ± 0.91	4.30 ± 1.03	4.39 ± 0.92	4.45 ± 0.77

Item 1—Thinking about last the week, have you been happy at school? Item 2—Thinking about the last week, have you got on well at school? Item 3—Thinking about the last week, have you been able to pay attention? Item 4—Thinking about the last week, have you got along well with your teachers?

**Table 6 healthcare-10-01198-t006:** The internal consistency and test–retest reliability analysis.

Area of Investigation (N Items)	IC Cronbach ‘s Alpha			Re-Test ICC		
	Phase 1—Nov. 2018	Phase 2—May 2019	Phase 3—Oct. 2019	Phase 1—Nov. 2018	Phase 2—May 2019	Phase 3—Oct. 2019
Physical wellbeing (N = 5)	0.657	0.705	0.572	0.603	0.689	0.581
Psychological wellbeing (N = 7)	0.615	0.585	0.564	0.617	0.542	0.519
Autonomy and parent’s relationship (N = 7)	0.776	0.759	0.646	0.758	0.712	0.606
Peers and friends’ support (N = 4)	0.661	0.638	0.554	0.643	0.648	0.511
School environment (N = 4)	0.598	0.766	0.661	0.584	0.721	0.655

IC = internal consistency; ICC = intra-class correlation coefficient.

**Table 7 healthcare-10-01198-t007:** The effect sizes using Cohen’s d between Arad and Cluj-Napoca children.

No.	Area of Investigation (N Items)	Phase of Evaluation		
		Phase 1–Nov. 2018	Phase 2–May 2019	Phase 3–Oct. 2019
1	Physical wellbeing (N = 5)	0.14	0.21	0.19
2	Psychological wellbeing (N = 7)	0.20	0.17	0.11
3	Autonomy and parent’s relationship (N = 7)	0.49	0.56	0.62
4	Peers and friends’ support (N = 4)	0.61	0.53	0.50
5	School environment (N = 4)	0.69	0.73	0.71

**Table 8 healthcare-10-01198-t008:** Known-group validity test.

	η^2^	F-Value	*p*
Socioeconomic status	0.061	26.76	<0.001
Academic performance	0.067	29.35	<0.01
Health status	0.034	18.59	<0.001

**Table 9 healthcare-10-01198-t009:** Differences between Arad and Cluj-Napoca [48,49,50,51].

Parameter Descriptor	Arad	Cluj-Napoca
Population	159,074	324,576
Population density no/km^2^	3445	1808
Ethnics - Romanians (%) - Hungarians (%) - Romani (%) - Germans (%)	85.2 10.1 1.7 0.8	81.5 16.4 1.1 0.2
Forbes Best Romanian cities in 2019	4th	3rd
Unemployment rate (%)	1.4	1.8
Average salary (RON)	2398	3115
Gross domestic product (GDP) per capita (EUR)	8930	12,400
Universities	2	10
Access to transport - National roads - Motorways	3 (69, 7, 7E) 2 (A1 and A11)	3 (1, 1C, 1F) 2 (A3 and A10)
Passenger traffic in airports (2016)	0	1,880,319

## Data Availability

Not applicable.

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
