# Peer review of "The KIDSCREEN-27 Quality of Life Measure for Romanian Children Aged 6: Reliability and Validity of the Romanian Version"

_healthcare, 2022, doi:10.3390/healthcare10071198_

Round 1

Reviewer 1 Report

The manuscript address the validation and application of a questionnaire to determine the health-related quality of life to a population of 6 years of age in two cities in Romania, a region where there is limited information, particularly on the population studied.

The manuscript is well written, it is easy to understand, the methods used are well established and justified, and the results and discussion are adequate. The conclusion is in line with the results obtained and the objectives set out in the article.

I consider that the validation part of the instrument is well founded, and the results presented throughout the article are pertinent.

In my opinion, the article can be accepted in its current form.

Author Response

Thank you very much for your disponibility to review our work and for your appreciated report.

Reviewer 2 Report

The authors propose to validate the Kidscreen 27 quality of life tool in a new age group, 6yr old healthy children in 2 cities in Romania (provincial vs. developed).

The Introduction is difficult to follow; it would be more relevant to address the importance of quality of life norms in a healthy population  or findings in older groups of healthy Romanian children, rather than the references to mental health and obesity, which is not the focus of this study. 

The results appropriately address the findings related to validity and reliability of the instrument. The age range, median, standard deviation of the children should be included.

The results are confounded by the use of specialists who discussed the importance of physical activity during the study but not with all participants. The results section needs to be edited to remove any statements that attribute causation to correlations.  A limitation is timing of the study-- start of the school year vs. later in the year, could have an effect on the child's stress level at various data points. The conclusion needs to be edited to removed statements of causation vs. correlation. 

Author Response

  • Thank you very much for your disponibility to review our work and for your appreciated report.
  • The Introduction has been redone, please verify. All the changes in the manuscript are evidenced in red.

  • As you suggested, it was included in the results the age range, median and standard deviation of the children (page 3, line 44-46). Please verify the text marked in red.

  • Thank you very much for your valuable observation; the importance of physical activity is more related to the age and to the BMI of children. This is the reason why, we have added a few more details (page 14, line 25-31). Please verify the text marked in red.

  • We completely agree with this pertinent observation. The change of school level, from kindergarten to preparatory class and later to the primary education has a great impact on the child’s mood and stress. This is why, we have added your valuable observation in our manuscript (page 17, line 46-50).

Reviewer 3 Report

The authors present the study on the use of The KIDSCREEN-27 quality of life measure tool in the Romanian pediatric population. The study has been well designed and with important results to be able to carry out the study with this tool in a reliable way in Romanian population.

However, there are some points that are important to highlight:

major changes

1. It is highly recommended to justify why the study was carried out in such a young child population, since in the other studies; as the authors themselves mention; has been performed in populations older than 8 years. Discuss advantages and disadvantages

2. The study, even though it presents important results, does not compare its results with previous studies. Are there previous reports on the Romanian-speaking population? As far as I know, there is the article Quality of Life in Romanian Children with Type 1 Diabetes: A Cross-Sectional Survey Using an Interdisciplinary Healthcare Intervention, and with which the results should be compared as it is in a population of the same nation. On the other hand, what differences or similarities in results exist in terms of other studies.

3. The conclusions at the end go around it is necessary to monitor the health status in the child population, when in reality it should be that the tool was reliable and valid for the evaluation of the population studied

minor changes

3. Even when the authorization number of the research ethics committee is included in the Institutional Review Board Statement, it would be advisable to note it in the materials and methods section.

4. The English must be reviewed by the native speaker since there are some small details in the wording

Author Response

  • Thank you very much for your disponibility to revise our work and also for you appreciated and valuable comments/observations.
  • You have right. We should have detailed why we chose such small children in the study, but, at your suggestion, we justified the motivation for the choice. Please verify the text marked in red at page 6, line 8-32.

  • As far as we know, no study has been published in the literature regarding the quality of life of 6-year-old Romanian children. In the article you mentioned, it is about Romanian children, but who suffer from type 1 diabetes, aged between 7 and 17 years. We quoted that study in the Introduction, but, at your suggestion, we tried to make a correlation between the two studies at the Discussion section, as well as the differences and similarities in results between our study and the studies of other researchers. Please check the text marked in red from the Discussion section.

  • The conclusion section at the end, was rethought. Please verify the text marked in red.

  • The authorization number given by the ethics committee of the faculty was noted also in the Materials and Methods section (subsection 2.2. Ethics statements).

  • The English was reviewed by a native speaker, please verify the entire text.

Round 2

Reviewer 3 Report

The authors have made the suggested changes, which has improved the quality of the article. Only minor changes to the English would be necessary.

Author Response

Dear Academic Editor, thank you very much for your valuable and very appreciated observation. We had no idea about the False Discovery Rate, but, we researched, we did the statistical calculations and that is a gain for us, thanks to you. According to the procedure described step-by-step by Cramer et al., the obtained adjusted p-values regarding our study, are still significant. Please let us know if you would like to replace the normal p values with the adjusted p values.

            We inserted in the text, at the statistics section (2.4.), that for a better interpretation of the obtained results, we performed the False Discovery Rate (FDR), to increase the power of the test. Likewise, the last paragraph before the discussion section briefly describes the results obtained.